# Analytical Methods for the Characterization of Vegetable Oils

**DOI:** 10.3390/molecules28010153

**Published:** 2022-12-24

**Authors:** Agnese Giacomino, Paolo Inaudi, Gessica Silletta, Aleandro Diana, Stefano Bertinetti, Elisa Gaggero, Mery Malandrino, Federico Stilo, Ornella Abollino

**Affiliations:** 1Department of Drug Science and Technology, University of Torino, 10125 Torino, Italy; 2Department of Chemistry, University of Torino, 10125 Torino, Italy

**Keywords:** authentication, chemometric treatments, EVOO, marker, metals, redox profile

## Abstract

The determination of the authenticity of extra virgin olive oils (EVOOs) has become more interesting in recent years. Italy is the first country in Europe in terms of number of Protected Designation of Origin (PDO) oils, which connects consumers to a feeling of tradition and thus to higher quality standards. This work focused on the consideration of the inorganic content as a possible marker of EVOOs. Ten vegetable oils (VOs), eight Italian EVOOs and seven not Italian EVOOs were analyzed. After pretreatment by acid mineralization, Al, Ba, Ca, Cu, Fe, K, Li, Mg, Mn, Na, P, Sb, Se and Zn were determined by ICP-OES. The electrochemical properties of a selected group of EVOOs and other vegetal oils of different botanical origin were investigated by voltammetry. Carbon paste electrodes (CPEs) were prepared. The features observed in the voltammograms reflect the reactions of electroactive compounds, which are present in the oils. A chemometric treatment of the results was performed to assess the possibility to distinguish (i) the region of provenience considering the inorganic profile; and (ii) the plant species from which each oil was obtained on the basis of the current profile registered during voltammetric analysis. Inorganic composition seems to be a useful marker for the assessment of the geographical origin of an EVOO. The EVOO-CPEs voltammetry seems to have a good ability to distinguish the plant species of origin. This method could be useful to monitor the conservation status of the oils, as the redox profile is linked to the oxidative degradation state.

## 1. Introduction

In the context of the global market, the certification of food quality is one of the most important goals for scientists in the agri-food sector. The determination of the authenticity and traceability of food products is of great interest for consumers, producers and distributors, and for these reasons international agencies have published specific guidelines on quality standards for oils [1].

Vegetable oils (VOs) and extra virgin olive oils (EVOOs) are widely used in the cooking and alimentary, cosmetic, pharmaceutical and chemical industries [2]. The EVOO is considered essential for the Mediterranean diet, and for this reason it has been extensively investigated in an effort to identify its geographical origin and detect frauds and adulteration [3,4,5,6].

The European Union (EU) has addressed the regulation, production and commercialization of oils. EU Regulation No. 1019/02 defines how to correctly pack and label oils, and the Commission Implementing Regulation EU No. 1335/13 (2013) made it obligatory to indicate the geographic origin on the label. According to EU Regulation No. 29/12 (European Commission Implementing Regulation, 2012), information concerning the geographic area in which olives are harvested and olive oil is obtained should be stated on the packaging or the label. For greater clarity, the document also defines that simple provisions such as ‘blend of olive oils of European Union origin’ or ‘blend of olive oils not of European Union origin’ or ‘blend of olive oils of European Union origin and not of European Union origin’ should be stated for the labeling of the origin of the oils. 

Olive trees are widely cultivated throughout the Mediterranean area and give rise to products of prominent peculiarities and quality when compared with products in other parts of the world. The European Union is the leading producer, consumer and exporter of olive oil. On average, the EU has produced 68.4%, consumed 54.2%, and exported 66.9 % of the world’s olive oil since 2013 [7]. In particular, Italy plays an important role in the production of high-quality oil [8], also differentiated by region of origin. Specifically, more fruity flavors and a fairly pronounced taste are associated with oils of the northern part of Italy, while a fuller and more strong character is more typical of ones from the south [9]. 

The olive tree belongs to the *Oleaceae* family, and thus it is an evergreen tree. It is a plant typical of the Mediterranean flora, but is able to adapt to temperate climates at other latitudes as well. *Olea Europaea L.* has many subspecies, however the distinction of particular economic importance is the one between the wild varieties (*Olea Europaea Var. Sylvestris (Mill.) lehr*), characterized by small fruits containing little oil, and the cultivated variety (*Olea Europaea Var. Europaea L.*), which produces edible fruits with characteristics that vary greatly depending on the cultivar, the soil, and the climate conditions in which the plant has grown.

Due to the physiological activity of the cells during the ripening of the fruit, which can occur in different periods of the year depending on the cultivar, the chemical composition of the drupe may be modified both for the organic fraction (fatty acids and minor components) and the inorganic (e.g., more magnesium and phosphorus, less calcium) one. With maturation, in fact, the number of water-soluble substances is reduced in favor of the lipid component: this process implies an increasingly critical state for the preservation of olive oil and consequently for its future quality. The oil is, in fact, predominantly contained in vacuolar structures within the cell, well separated from the cytoplasmic enzymes. In cases of excessive maturation, vacuolar barriers degrade and allow the contact of lipase and lipoxidase with the lipid content of these vacuoles. Due to the activity of these two enzymes, the inactivation of triglycerides (by action of lipase) and rancidity (due to the intervention of lipoxidase) may occur [10,11].

For this reason, the time of harvest of olives greatly influences the quality of the oil produced: it is customary to consider an oil obtained from overripe olives to be of inferior quality, whereas olives that are too immature will have a more aggressive taste of bitterness and spiciness [12]. 

Adulteration is a serious issue for EVOOs, and it may take place by mixing EVOO with oils of lower quality, for example by soybean oil [13]. Many analytical techniques have been used to assess the quality and investigate the adulteration of oils, for example 1H and 13C nuclear magnetic resonance (NMR) imaging [14], near infrared resonance (NIR) [15] and mid infrared resonance (MIR). Some other spectroscopic techniques, such as Raman spectroscopy [16,17,18], fluorescence spectroscopy [19,20,21], Fourier transform infrared (FTIR) spectroscopy [22,23], and ultraviolet spectrometry [24,25] have great potential for characterizing different olive oil samples from the point of view of organic substances and hence detecting their adulteration. Inductive coupled plasma optical emission spectroscopy (ICP OES), inductive coupled plasma mass spectrometry (ICP MS) and graphite furnace atomic absorption spectrometry (GF AAS) [26,27,28,29,30] are widely used to investigate the trace element content in oils. 

In addition, high performance liquid chromatography (HPLC) and gas chromatography (GC) [31,32,33,34,35,36,37], frequently coupled with MS, were employed to detect EVOOs adulteration concerning organic components. Several publications have described the use of volatile-species distribution as a fingerprint to assess traceability, authentication, and non-degradation based on head-space sampling and GC. Most of these techniques are expensive and/or complex; thus, a simpler, more rapid, and cost-effective procedure [38] is highly desirable. Voltammetry fulfils these requirements and may represent a suitable alternative for detecting adulteration in olive oil. In two works, Zappi et al. [39,40] suggested that correlations between the electrochemical response of a modified screen-printed electrode and the olives cultivar and/or their geographical origin could be established.

The objective of the present manuscript are twofold. The first one is to investigate the relationship between the inorganic components of oils, determined by ICP-OES, and their geographical origin, matching the definition of an “inorganic fingerprint” characterizing each oil. Metals present in oils mainly derive from soils in which olive trees were grown, but may have other sources: atmospheric fallout, fertilizers, and metal-containing pesticides, or contamination from the metal-based processing equipment [27,41].

The second aim is to investigate the oxidation and reduction profile of different oils [42] related to the presence of antioxidant compounds, such as polyphenols, tocopherols, carotenoids and chlorophylls using voltammetry. Analyses were performed both by cyclic voltammetry (CV) and square wave voltammetry (SWV) using modified carbon paste electrodes (CPE) in which the paste binder was made of the oil under consideration. The purpose is to examine the possibility of discriminating against the botanical origin of the different oils examined based on their content in terms of antioxidant compounds.

The results of chemical analysis were processed with principal components analysis (PCA) [43] to search for a fingerprint of origin of oils, resulting from the trace element determination and voltammetric analysis.

## 2. Materials and Methods

### 2.1. Samples

A total of 15 EVOO samples were investigated. Table 1 lists their codes, origins and classification: among these samples there are some monovarietal productions (indicated with MV), some Protected Designation of Origin (PDO) specimens, and some blends (mix of olives of different varieties). Eight samples come from two different regions of Italy: two from Liguria and six from Umbria. All the samples analyzed were produced and harvested in 2018–2019. Another seven oils of non-Italian production were analyzed: one from Croatia, four from Spain, and two from Portugal.

Ten samples of other vegetable oils (VOs), namely sunflower seed oil, corn seed oil, peanut seed oil, almond oil and sesame seed oil, purchased from a supermarket, have also been analyzed to compare their composition, in terms of redox properties and inorganic elements, with the composition of EVOOs. Samples are listed in Table 2.

### 2.2. Apparatus and Reagents

Sample dissolution was performed in tetrafluoromethoxyl (TFM) vessels, with a Milestone MLS-1200 Mega (Milestone, Sorisole, Italy) microwave laboratory unit.

Analyses of element contents were carried out with an inductively coupled plasma-optical emission spectrometer (ICP-OES), in particular the Perkin Elmer Optima 7000 (Perkin Elmer, Norwalk, CT, USA).

Analytical grade reagents were used throughout. Standard metal solutions were prepared from concentrated Merck TraceCERT stock solutions (Merck, Darmstadt, Germany).

Electrochemical analyses were made with an Autolab PGSTAT 10 analyzer (Eco-Chemie Metrohm, Utrecht, The Netherlands) connected to a Metrohm 663 VA Stand. The potentiostat is connected to a computer that allows the setting of conditions measurement, the display of voltammograms and the related data through the IME 663 interface and the GPES software (General Purpose Electrochemical System).

High purity water (HPW) obtained from a Milli-Q apparatus (Millipore, Bedford, MA, USA) was used throughout for the preparation of sample and standard solutions.

### 2.3. Procedures

#### 2.3.1. Profile of Inorganic Component

Unfortunately, no Standard Reference Materials (SRM) for trace elements in oils is available on the market. For this reason, SRM 1573a, tomato leaves, supplied by the National Institute of Standards and Technology (NIST, Gaithersburg, Maryland, United State) were analyzed to value the efficiency and the accuracy of analytes quantification. This SRM is primarily intended for use in botanical materials, agricultural food products, and materials of similar matrix.

It is very important to homogenize oil samples before the pretreatment, since the oil matrix tends to stratify, especially when there is a residue at the bottom of the container and/or if the oil has been subjected to thermal stress (stored in the freezer and then thawed for analysis). Acid digestion in the microwave oven was adopted to mineralize the samples, according to our previous work on similar matrices [44,45]. The stirring step is indispensable to homogenize each sample. 

Aliquots of 0.5 g of each sample were weighed directly in TFM vessels, then 3 mL of HNO_3_ 65% and 3 mL of H_2_O_2_ 30% were added. The digestion sequence (1 min–250 W, 1 min–0 W, 5 min–250 W, 5 min–400 W, 5 min–600 W, 20 min of ventilation) was applied through an external microprocessor, and then the vessels were left under a fume hood for two hours for a final cooling phase.

The sample solutions so obtained were analyzed by ICP-OES for a quantitative evaluation of Al, Ba, Ca, Cu, Fe, K, Li, Mg, Mn, Na, P, Sb, Se and Zn. For each analyte the optimal wavelength for signal measurement was chosen to obtain maximum response sensitivity and at the same time minimum interference by other species. The quantification of the elements was carried out following the external calibration method.

#### 2.3.2. Reductive Oxide Profile

CPEs were prepared by mixing graphite powder and the oil under investigation. Graphite and oil were previously weighed and homogenized. The obtained paste was then packed in the body of the electrode, represented by a polypropylene tube obtained from a syringe, after cutting its tip, and an electric contact, namely a copper wire, was inserted. The syringe plunger was put in contact with the paste, after removing its tip, and was used as a piston to remove the outer part of the paste after the analysis. Figure 1a–e shows the structure of the CPE.

After packing, the paste was smoothed by rubbing its outer surface on a sheet of weighing paper. Generally, the CPEs so obtained were used as prepared.

For each test sample, more than one CPE was prepared by varying the ratio between graphite powder and oil: 80:20, 70:30, 60:40 and 50:50, respectively. Each CPE was tested as a working electrode (WE) in an electrochemical cell filled with 0.1 M HCl as the supporting electrolyte. 

The best choice for the analysis was found to be the graphite/binder mixture in a 70:30 ratio, since both the voltammograms obtained by cyclic voltammetry (CV) and square wave voltammetry (SWV) show the most defined peaks and have background currents close to zero. Thus, this ratio was used throughout the study.

All samples were analyzed by CV. The scanning parameters set for the analysis were: potential range: 0–1, 15–0 V, scan: linear scan, scan rate: 0.1 V/s.

For all samples, a SWV analysis was also performed, scanning the potential in anodic and cathodic direction. This type of scan, using a pulsed waveform, generally allows for the obtaining of more intense signals and consequently more defined peaks than a linear scan. The parameters set in this case were: potential range: 0–1.15 V for oxidation, 1.15–0 V for reduction, potential step: 0.007 V, wavelength amplitude: 0.010 V, frequency: 15 Hz.

### 2.4. Chemometric Treatment

A PCA was carried out with the aid of the XLSTAT4.4 software package (Addinsoft, Paris, France), a Microsoft Excel plug-in. Unscrambler X 10.2 (Camo Analytics, Oslo, Norway) was employed for auto-scaling the dataset and for substituting values below the limits of detection (LODs) with estimated values.

## 3. Results and Discussion

### 3.1. Inorganic Component

#### 3.1.1. Reference Material

Table 3 reports the results concerning the analysis of SRM 1573a, tomato leaves. Percentage recoveries from 75% to 101.5% were obtained.

The accuracy of the results can be considered satisfactory: for this reason, the same procedure was applied to the oil samples.

#### 3.1.2. EVOOs and other Vegetable Oil Samples

Table 4 a and b show the results of the analysis of the inorganic components in oil samples. The data show that Al is present in all samples in concentrations of 0.6–0.9 mg kg^−1^, with the exception of EVOO1, EVOO13, almonds, and sesame oils in which it exceeds 1 mg kg^−1^. Al is considered toxic for plants at concentrations between 2 and 5 mg kg^−1^, since it limits their development [46]. Ca is an essential element; it is the element with the highest concentration in all samples. Ca content in EVOOs varied from 4 to 16 mg kg^−1^, while higher concentrations are present in several vegetable oils: in particular, in sesame oil and in almond oil its concentration is 29 mg kg^−1^ and 37 mg kg^−1^, respectively. The concentration range for Fe is 0.45–1.18 mg kg^−1^ in the Italian and Croatian EVOOs, while it is below the detection limit in the other EVOOs and in most vegetal oils. K shows a high variability of concentration in EVOOs (0.06–17.4 mg kg^−1^); it is mainly present in Umbrian and Croatian EVOOs, where it exceeds 5 mg kg^−1^. Mg and Na are present in most samples at comparable concentrations of 0.9–3 mg kg^−1^, with the exception of a few samples with lower Na concentrations and SES in which 9.32 and 4.46 mg kg^−1^ of Mg and Na are found, respectively. Se is a trace element found in all samples, albeit in very variable concentrations, irrespective of the plant species of origin. Some elements, namely Ba, Cu, Li, Mn, P, Sb and Zn, are not reported, because they were always below the LOD of the method.

It can be concluded that the presence of essential elements such as Na, Mg and Ca was detected in all samples, regardless of the botanical origin of the oil.

Some differences can be observed: EVOOs from Liguria contain a higher content of Al and Na than Umbrian EVOOs, but they present lower concentrations of K; the amount of Fe is comparable. It can be assumed that this is due to the peculiarities of the growing area. 

Comparing Italian and non-Italian EVOOs, Fe is present at detectable levels in the former, while it is below the LOD in the latter, with the exception of Croatian EVOO that is produced on the Adriatic coast. 

Sesame oil generally presents the highest concentration of most of the elements, namely Al, K, Mg and Na, and also has a high Ca level. Almond oil presents a remarkably high content of Ca.

The metal content mainly depends on the conditions in which the olives were grown, namely the composition of soil and water. In particular, the differences are influenced by the pedoclimatic conditions of growth of the botanical species of origin. For example, Umbria is a region of central Italy surrounded by green hills and mountains, while Liguria is a northern region characterized by the presence of coasts for half of its borders; the relatively high concentrations of Na in Ligurian samples, in comparison with the other EVOOs, may be due to the contribution of marine spray. Furthermore, the conditions of production, storage and transport of oil may affect the element content [26].

### 3.2. Redox Profiles

#### 3.2.1. Method Development with Commercial EVOO

The redox profile of the samples was investigated. It is related to the presence of antioxidants in the oil matrix, capable of taking part in electron transfers typical of redox reactions. 

Each oil was mixed with the graphite powder, acting simultaneously as both binder and sample. The electrochemical response of each CPE reflects the oxide-reductive properties of electroactive compounds present in it, such as polyphenols, tocopherols, carotenoids and chlorophylls. CPEs were prepared just before use. In any case, a repeatability of the response was verified after three days, during which they were stored at room temperature in the dark and covered with Parafilm.

Figure 2 and Figure 3 show the voltammograms obtained by CV and SWV, respectively, using different oil: graphite ratios (20:80, 30:70, 40:60 and 50:50). Experiments were performed with an EVOO purchased in a local supermarket.

Overall, the most defined peaks are shown, both in CV and SWV, using the graphite/oil 70:30 ratio, so this ratio was used throughout the study.

In accordance with the literature [47], the observed peaks can be traced back to a given electroactive chemical family. Namely:-peaks around the potentials of 0.4 V and 0.6 V correspond to polyphenols-the peaks around the 1.1 V potential are relative to tocopherols.

#### 3.2.2. EVOO and VO Samples

The voltammograms acquired with SWV show more defined peaks than those obtained using CV, but the results of both techniques will be discussed hereafter, because they provide complimentary information. 

In EVOOs (Appendix A Appendix A) the redox profiles obtained with anodic scans show up to four peaks: as stated above, the peaks correspond to two chemical species, namely polyphenols at potential around 0.4 V, 0.6 V and 0.8 V and tocopherols at about 1 V. 

Cathodic and CV scans (Appendix A Appendix A) display from two to three peaks in the voltammograms relating to the EVOOs and from one to two peaks in those associated with other oils (Appendix A Appendix A). Specifically, for EVOOs there are one or two peaks between 0.4 V–0.6 V, and in more than half of the samples there is also a peak at around 1.1 V.

In VOs (Appendix A Appendix A) the peaks related to SWV-A are less numerous (up to three in Appendix A) and have lower height than in EVOOs. In all cases there is a low peak at 0.4 V (polyphenols), in sunflower and corn there are peaks around 0.6 V (polyphenols, Appendix A Appendix A) and in Appendix A there is a third peak around 1.1 V (tocopherols, Appendix A Appendix A).

In VOs, instead, there is always a peak at 0.4 V and in some samples a second peak appears: for CS1, CS2, PS2 (Appendix A Appendix A) and all corn seed oils it is found at about 1.1 V, while for sesame oil it is observed near 0.2 V (Appendix A Appendix A).

It is possible to notice some differences between the profiles of the EVOOs and those of VOs. The first one shows defined and higher peaks for tocopherols at potentials of about 0.6 and 1 V, and in some cases a small peak at 0.4 V. In the voltammograms of other oils it is possible to recognize a less sharp peak at about 0.6 V for sunflower, corn and peanut, and at about 0.4 V for sesame and almond. From this first comparison we can already assert that a peak related to polyphenols is present in all oils, independently of the botanical species of origin, while the tocopherols seem most present in the EVOOs. 

The results obtained by both CV and SWV suggest that EVOOs are richer in redox-active species, such as polyphenols and tocopherols, compared to VOs; therefore voltammetry confirms that the former represent a valuable source of antioxidants. 

Peaks are different in shape, position or intensity, being associated with antioxidant molecules of different natures and present in different concentrations in relation to the botanical species from which the oil comes.

#### 3.2.3. Applications of Redox Profiles

##### Identification of Possible Adulterations with Other Oils

To assess the applicability of the technique to check the presence of counterfeits, due to the adulteration of EVOOs with oils of different botanical origin, mixtures of EVOO and other vegetable oils in a ratio of 5:1 were prepared. CPEs were made with these mixtures, and SWV-A voltammograms were compared with those of the two oils of origin. 

Figure 4 shows, as an example, the comparison between the voltammograms obtained with EVOO1 (100%): CPE, SS1 (100%)-CPE and EVOO1:SS1 = 5:1 mixture-CPE.

It is possible to observe that a small amount of SS significantly changes the redox profile, thanks to the variation of the amount of the antioxidant species characteristic of each oil. These results show that the comparison of voltammograms can be used to detect the presence of an adulteration of EVOOs.

##### Assessment of the Conservation Status

An improper storage of oils can negatively affect their stability and consequently give rise to changes in chemical composition, which in turn can affect their organoleptic characteristics. To assess the applicability of the technique to monitor the conservation status of the oil, an aliquot of EVOO was exposed to direct sunlight in a closed vessel. Three CPEs were prepared: the first one using a sample stored in the dark (hereafter called “fresh EVOO”), and the second and third with the sample after one and two months of exposure, respectively. SWV-A voltammograms were recorded. The results are reported in Figure 5.

It is possible to observe a difference between the three samples. In particular, the peaks at 0.5 and 1.0 V decrease after two months of exposure to sunlight. The oxide-reduction profile could therefore provide informative data on the state of each oil in terms of both proper preservation, especially for relatively long periods, and organoleptic quality.

### 3.3. Chemometric Treatment

The experimental results were processed by Principal Component Analysis (PCA), one of the main multivariate chemometric techniques, which enables a multi-variable reality to be represented with a few variables, namely the PCs. The latter are obtained by linear combinations of the original variables and are orthogonal to each other. PCs maintain the total variance of the original variables but change its distribution: most of the variance is retained in the first two or three PCs. PCA allows to reduce the size of data, represent experimental data in an orthogonal space, eliminate spurious information such as instrumental background noise, assess the relative relevance of the variables, and visualize the samples graphically allowing for the search for classes, clusters and outliers. PCA plots rely on two graphical elements: scores, i.e., points representing samples in analysis; and loadings, i.e., vectors with origin at the intersection of the axes representing the variables. PCA loadings are the coefficients of the linear combination of the original variables from which the principal components (PCs) are constructed. These elements can be represented individually (loading and score plots) or in combination (biplots). 

In this way, it is possible to evaluate the degree of similarity of the different samples on the basis of investigated variables, which in this work are the concentrations of the elements and the current intensities measured during the potential scans.

#### 3.3.1. PCA of the Inorganic Components

The biplot for all of the oils (Figure 6A) shows how oils with similar origin are closer together in the graphic representation. Concerning EVOOs (Figure 6B), Portuguese EVOOs (highlighted in blue in the Figure 6B) are very close to each other, their great similarity deriving from the fact that they were supplied by the same producer that cultivates in the same area; they are characterized by the important content of Se. The other EVOOs tend to be grouped in a less defined area. The Spanish EVOOs are grouped in the same quadrant, in correspondence to relatively high concentrations of Ca, Al and Mg, with the exception of EVOO10, which is closer to the Portuguese samples. This behavior could be explained by the fact that it has a greater concentration of Se and that it is a single variety PDO (like the Portuguese), while the other Spanish samples are made from blends of different varieties of olive. The Umbrian EVOOs are also quite close to one another, around the loadings of K and Fe, but occupy a much larger area in the plot; an exception is given to EVOO3, which is a bit further away from all other samples owing to the greater quantity of Se. The Ligurian EVOOs, on the other hand, are both towards the right end of the chart, although not being adjacent to each other, because of the different quantities of Al and Mg. It should be noted that the Italian EVOOs are in the upper part of the chart, except for EVOO3 and EVOO1, which, however, are very close to the axis, so they are characterized by a greater presence of K, Fe and Na than the other EVOOs. Furthermore, EVOO9 of Croatia is located in the upper part of the plot; it is possible to assume that this is due to its geographical origin, since the company that produces it is located on the Adriatic coast, a few dozen kilometers away from the Italian one. On the contrary, non-Italian EVOOs, being at the bottom of the graph, are characterized by a greater similarity related to the concentrations of Se, Ca, Al and Mg.

#### 3.3.2. PCA of CV and SWV Results

As regards the processing of data obtained by voltammetry measurements, a matrix with the potential values and the corresponding current values obtained experimentally for each sample was constructed. The variables are represented by the potential values, and the scores are obtained as linear combinations of the current values at different potentials. The chemometric treatment was applied to the results obtained both by CV (Figure 7) and by SWV (Figure 8 and Figure 9).

From the observation of these graphs, it is possible to note the presence of a good separation between the samples of EVOO and those of other oils that was expected from the remarks made in Section 3.2: in fact, it seems that the EVOOs occupy a more defined region of the vector space, while the oils of other nature are positioned in more diffused way around the EVOOs. In PC graphs obtained from the results of SWV-A, SWV-C and CV, Ligurian oils (circled in light green) tend to distance themselves from the other EVOOs, but always occupy a region of the vector space opposite to the oils of different botanical origin. In conclusion, there is the possibility of successfully discriminating not only EVOOs from oils of other natures, but also Ligurian EVOOs from Umbrian ones instead, whereas it is not so easy to distinguish Portuguese from Spanish EVOOs.

The method can therefore be useful for a botanical and merceological characterization.

## 4. Conclusions

The first part of this work was aimed at characterizing EVOOs samples from the point of view of the inorganic profile. The content of fourteen elements, namely Al, Ba, Ca, Cu, Fe, K, Li, Mn, Mg, Na, P, Sb, Se and Zn was quantified by ICP-OES

Ca, Mg, Na and Se are the major elements (>1 mg kg^−1^). K, on the other hand, had a very variable trend. The data show that Italian EVOOs have greater Fe content than non-Italian ones, and Umbrian EVOOs show higher K but lower Na Al concentrations than Ligurian ones. The composition of the oils obtained from other plants is rather heterogeneous and no common trend can be identified.

The chemometric treatment of the experimental results via a PCA confirmed the possibilities offered by the method with regard to the discrimination of geographical origin on the basis of the proximity of the different scores, although some samples failed to meet expectations because of a higher concentration of Se. In particular, the behaviour of the two samples from Portugal shows the possibility to distinguish oils provided by different producers if the soil and the other characteristics of the area cultivated by each producer differ from each other. In synthesis, a distinction at local scale is feasible.

A geographic and varietal characterization is therefore possible even if the analysis is critical because of the complex organic matrix. Thus, the elementary analysis of oils can be a means to evaluate product authenticity, for example by detecting cuts with oils of different botanical species, or to determine the pollution from heavy metals derived from the soil in which olive trees were grown or from accidental contamination during the extraction or distribution phases. Hence, it can be a useful tool for monitoring the quality (from the point of view of contamination by metals) and the genuineness (understood as absence of sophistication) of the oil itself.

The presence of species with antioxidant properties, such as polyphenols and tocopherols, was detected from the peaks displayed in the voltammograms. SW provides more intense peaks than CV. The peaks in the profiles of different oils are different in shape, position, or intensity, because they are associated with antioxidant molecules of a different nature (even if they share common features, i.e., the range of potential in which they appear) and are present in different concentrations, which in turn are associated with different compositions of the species from which the oil derives. Therefore, voltammetry allowed us to discriminate different types of oils according to their botanical origin, both through the simple observation of the voltammetric profiles, and by the chemometric treatment of the results by PCA.

Chemometric treatment with PCA confirmed that the different oils are separated quite well in the score plot according to their botanical origin and in some cases also for their geographic origin. In fact, not only the Spanish EVOOs were distinguished from the Italian ones, but the Umbrian samples were also differentiated from the Ligurian ones, while the separation between Spanish and Portuguese EVOO was not as efficient. 

As expected, non-EVOO samples were differentiated according to the plant species of origin.

In turn, these differences in chemical composition affect both the organoleptic characteristics and the stability of the oils. The antioxidant molecules, in fact, can take part in oxidation reactions to which the vegetable oils are naturally subject, protecting them from lipoperoxidation but also giving a bitter and pungent taste to the oil. A high content in metals involves a catalysis of the lipoperoxidation reactions that cause the early rancidity of the oil.

In this sense, voltammetry can be used as a monitoring tool for the production and distribution chain of oil and derivatives, assessing their conservation status, as well as for the identification of possible sophistication with oils of different botanical origins that would vary the intensity and the position of the peaks proper to the EVOO.

In conclusion, voltammetry presents good potential from the standpoint of monitoring the quality, state of conservation and the organoleptic properties of oils and their by-products. Possible future directions resulting from study includes, for the inorganic profile, the analysis of both oils and the soil in which the olives were grown; and for the redox profile, the analysis of a higher number of samples to better investigate the potential of the technique.

## Figures and Tables

**Figure 1 molecules-28-00153-f001:**
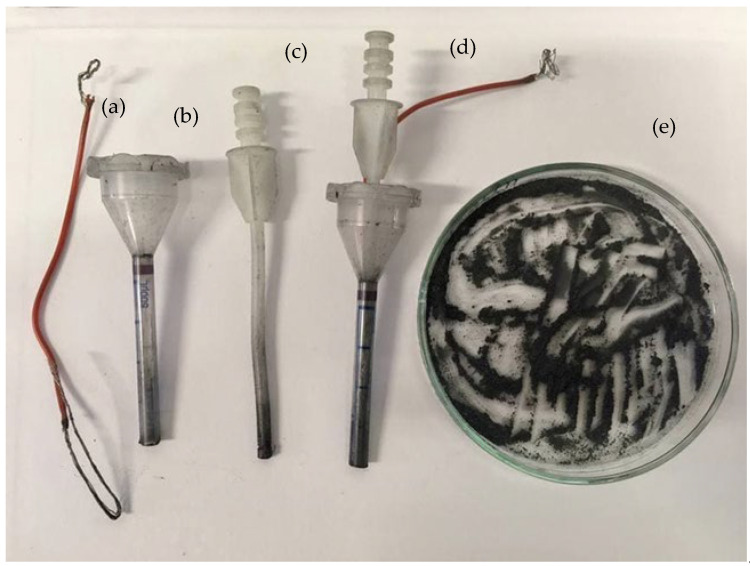
Structure of the CPE (**a**) copper wire, (**b**) polypropylene tube, (**c**) syringe pluger, (**d**) complete CPE, (**e**) paste/oil mix.

**Figure 2 molecules-28-00153-f002:**
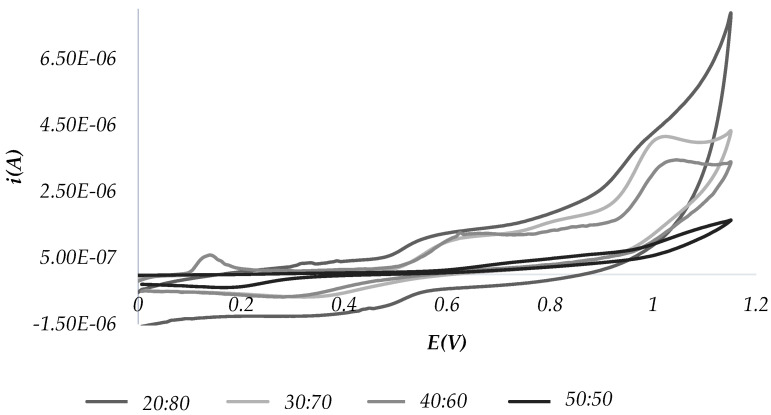
CV scans recorded, varying the oil: graphite ratio in the CPE.

**Figure 3 molecules-28-00153-f003:**
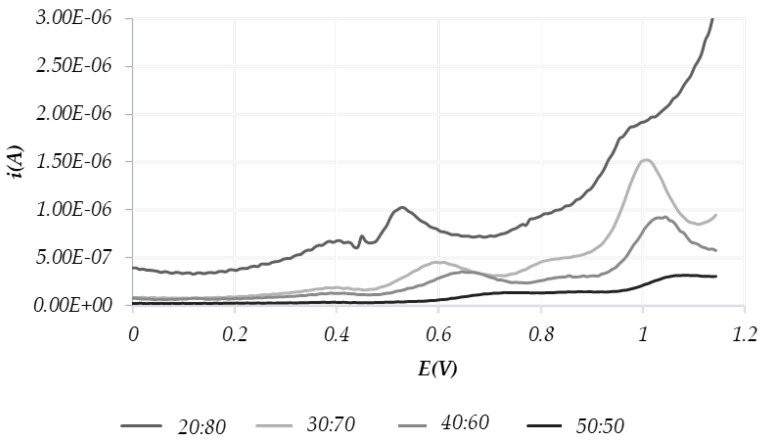
SWV scans recorded varying the oil:graphite ratio in the CPE.

**Figure 4 molecules-28-00153-f004:**
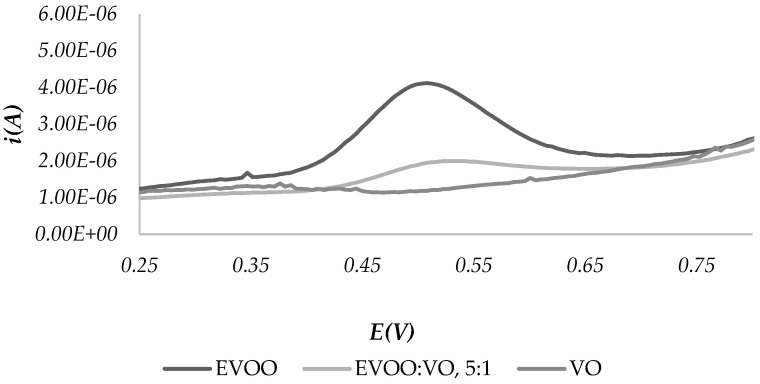
SWV-A scans for the comparison of a mixture of EVOO and SSO with the pure oils.

**Figure 5 molecules-28-00153-f005:**
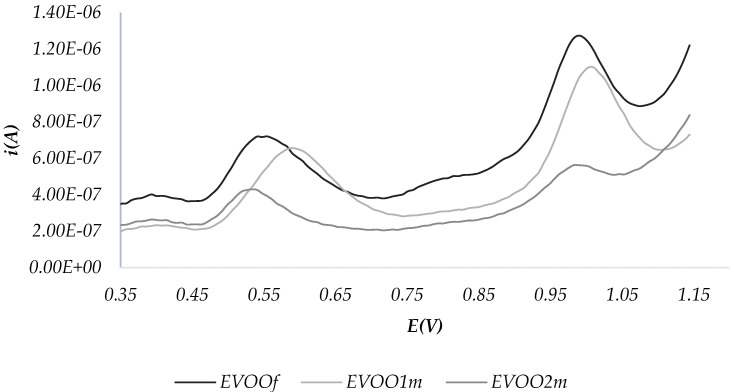
SWV recorded for fresh EVOO (EVOOf), after one (EVOO1m) and two (EVOO2m) months of exposure to sunlight.

**Figure 6 molecules-28-00153-f006:**
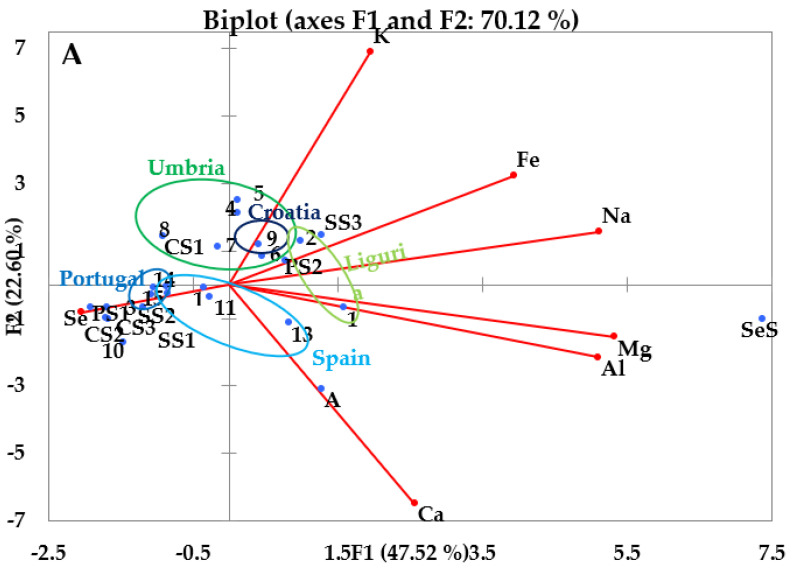
(**A**) PCA of inorganic component in all the investigated oils; (**B**) PCA of inorganic component in EVOOs. EVOO1, 2: Liguria, EVOO 3-8: Umbria, EVOO9: Croatian, EVOO10, 11, 12, 13: Spain, EVOO14, 15: Portugal. VOs–SS1: Sunflower Seed Oil (Brand A), SS2: Sunflower Seed Oil (Brand B), SS3: Sunflower Seed Oil (Brand C), CS1: Corn Seed Oil (Brand A), CS2: Corn Seed Oil (Brand B), CS3: Corn Seed Oil (Brand D), PS1: Peanut Seed Oil (Brand A), PS2: Peanut Seed Oil (Brand E), A: Almond Oil (Brand F), SeS: Sesame Seed Oil (Brand G).

**Figure 7 molecules-28-00153-f007:**
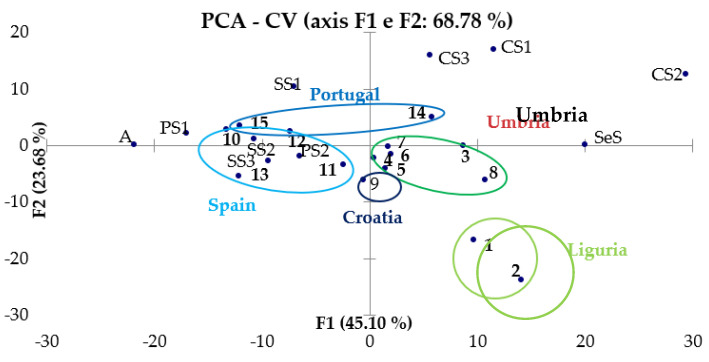
PCA of CV data.

**Figure 8 molecules-28-00153-f008:**
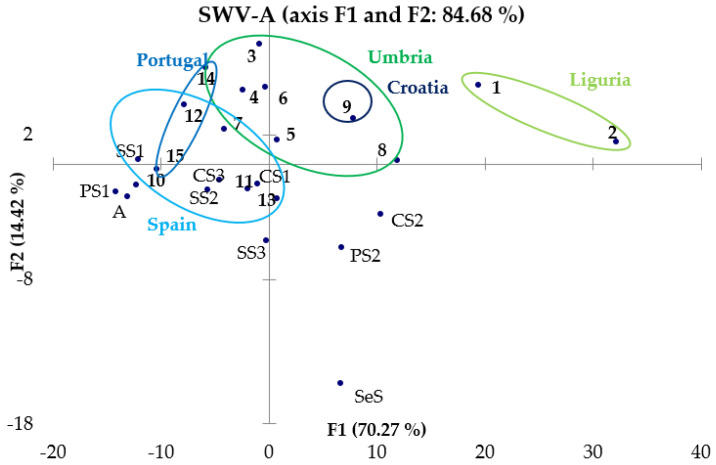
PCA of SWV-A data.

**Figure 9 molecules-28-00153-f009:**
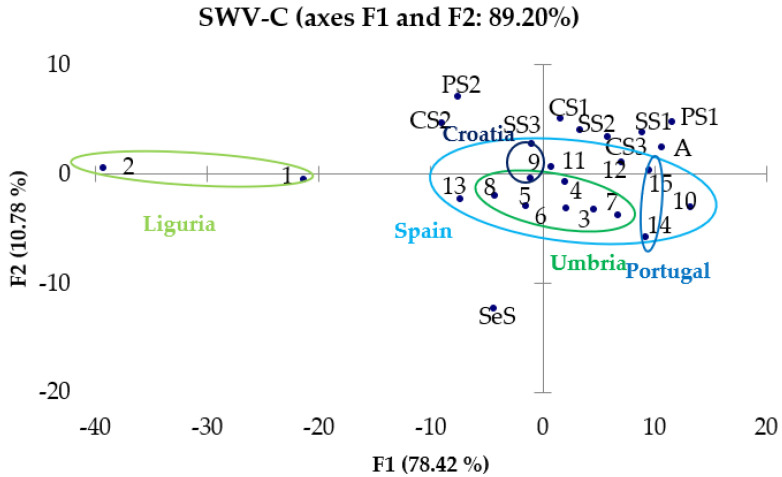
PCA of SWV-C data.

**Table 1 molecules-28-00153-t001:** List of EVOOs samples.

Code	Region	Classification
EVOO1	Liguria	MV
EVOO2	Liguria	MV
EVOO3	Umbria	PDO
EVOO4	Umbria	PDO
EVOO5	Umbria	Blend
EVOO6	Umbria	Blend
EVOO7	Umbria	Blend
EVOO8	Umbria	Blend
EVOO9	Croatian	MV
EVOO10	Spain	PDO
EVOO11	Spain	MV
EVOO12	Spain	MV
EVOO13	Spain	MV
EVOO14	Portugal	PDO
EVOO15	Portugal	PDO

MV: monovarietal, PDO: Protected Designation of Origin, Blend: mix of olives of different varieties.

**Table 2 molecules-28-00153-t002:** VO samples.

**Code**	**Classification**
SS1	Sunflower Seed Oil (Brand A)
SS2	Sunflower Seed Oil (Brand B)
SS3	Sunflower Seed Oil (Brand C)
CS1	Corn Seed Oil (Brand A)
CS2	Corn Seed Oil (Brand B)
CS3	Corn Seed Oil (Brand D)
PS1	Peanut Seed Oil (Brand A)
PS2	Peanut Seed Oil (Brand E)
A	Almond Oil (Brand F)
SeS	Sesame Seed Oil (Brand G)

**Table 3 molecules-28-00153-t003:** Results of the analysis of SRM—Tomato Leaves (mg kg^−1^).

Element	Certified Value	Experimental Result	Recovery (%)
Al	598 ± 12	451 ± 15	75.5
B	33.3 ± 0.7	27.5 ± 0.3	82.7
Ca	50,500 ± 900	45,072 ± 1604	89.2
Cd	1.52 ± 0.04	1.32 ± 0.01	87.1
Co	0.57 ± 0.02	0.64 ± 0.01	112
Cr	1.99 ± 0.06	1.87 ± 0.02	94.2
Cu	4.70 ± 0.14	4.05 ± 0.09	86.2
Fe	368 ± 7	284 ± 4	77.2
K	27,000 ± 500	24,455 ± 916	90.6
Mg	12000 *	9706 ± 307	80.9
Mn	246 ± 8	197 ± 5	80.1
P	2160 ± 40	2192 ± 28	101.5
Sr	85 *	67.1 ± 0.8	79.0
Zn	30.9 ± 0.7	27.0 ± 0.4	87.4

*informative value.

**Table 4 molecules-28-00153-t004:** Inorganic element content (mg kg^−1^) in (a) EVOOs and (b) VOs.

EVOO							
Sample	Al	Ca	Fe	K	Mg	Na	Se
**EVOO 1**	1.22 ± 0.03	11.4 ± 0.3	0.59 ± 0.02	0.43 ± 0.06	2.90 ± 0.04	2.59 ± 0.09	1.42 ± 0.03
**EVOO 2**	0.79 ± 0.02	9.32 ± 0.30	0.99 ± 0.02	9.97 ± 0.72	2.08 ± 0.02	2.80 ± 0.07	1.47 ± 0.02
**EVOO 3**	0.65 ± 0.02	16.2 ± 0.4	0.94 ± 0.02	5.53 ± 0.15	1.86 ± 0.02	1.07 ± 0.08	4.17 ± 0.02
**EVOO 4**	0.68 ± 0.02	5.59 ± 0.18	0.56 ± 0.03	15.9 ± 0.02	2.11 ± 0.11	1.69 ± 0.10	1.25 ± 0.08
**EVOO 5**	0.67 ± 0.02	4.72 ± 0.19	0.70 ± 0.02	17.4 ± 0.5	1.56 ± 0.02	1.75 ± 0.07	0.95 ± 0.06
**EVOO 6**	0.94 ± 0.02	8.36 ± 0.61	0.47 ± 0.02	9.91 ± 0.13	2.26 ± 0.03	1.44 ± 0.11	1.10 ± 0.02
**EVOO 7**	0.83 ± 0.02	6.99 ± 0.10	0.49 ± 0.02	11.5 ± 0.2	2.17 ± 0.05	1.00 ± 0.02	1.78 ± 0.11
**EVOO 8**	0.60 ± 0.02	4.02 ± 0.18	0.89 ± 0.02	8.55 ± 0.22	1.47 ± 0.02	1.15 ± 0.06	2.21 ± 0.05
**EVOO 9**	0.75 ± 0.02	7.74 ± 0.60	1.18 ± 0.02	8.33 ± 0.29	2.07 ± 0.07	1.91 ± 0.03	1.43 ± 0.03
**EVOO 10**	0.60 ± 0.02	14.6 ± 0.3	<0.02	0.06 ± 0.06	1.54 ± 0.02	0.90 ± 0.15	3.61 ± 0.10
**EVOO 11**	0.70 ± 0.02	9.78 ± 0.13	<0.02	2.02 ± 0.05	1.90 ± 0.06	1.94 ± 0.02	1.16 ± 0.08
**EVOO 12**	0.72 ± 0.02	11.1 ± 0.3	<0.02	1.07 ± 0.03	2.03 ± 0.02	1.86 ± 0.17	1.09 ± 0.11
**EVOO 13**	1.21 ± 0.03	13.3 ± 0.7	<0.02	0.89 ± 0.06	2.75 ± 0.02	1.43 ± 0.04	1.08 ± 0.29
**EVOO 14**	0.63 ± 0.02	8.66 ± 0.30	<0.02	0.35 ± 0.02	1.82 ± 0.03	1.38 ± 0.28	1.90 ± 0.17
**EVOO 15**	0.64 ± 0.02	5.98 ± 0.21	<0.02	0.42 ± 0.02	1.95 ± 0.02	1.40 ± 0.08	1.79 ± 0.02
**MIN**	0.60	4.02	<0.02	0.06	1.47	0.90	0.95
**MAX**	1.22	16.2	1.18	17.4	2.90	2.80	4.17
**MEAN**	0.77	9.19	0.46	6.15	2.03	1.62	1.76
**VO**							
**Sample**	**Al**	**Ca**	**Fe**	**K**	**Mg**	**Na**	**Se**
**CS1**	0.60 ± 0.02	6.73 ± 0.18	<0.02	0.71 ± 0.02	2.22 ± 0.02	1.61 ± 0.19	1.58 ± 0.03
**CS2**	0.61 ± 0.03	13.9 ± 0.5	<0.02	<0.02	1.39 ± 0.02	0.24 ± 0.04	2.00 ± 0.08
**CS3**	0.66 ± 0.02	9.91 ± 0.81	<0.02	<0.02	1.15 ± 0.02	<0.02	1.61 ± 0.08
**SS1**	0.55 ± 0.02	12.3 ± 0.1	<0.02	0.03 ± 0.02	1.55 ± 0.07	0.24 ± 0.18	1.36 ± 0.02
**SS2**	0.64 ± 0.02	12.1 ± 0.5	<0.02	<0.02	1.56 ± 0.05	0.79 ± 0.08	1.17 ± 0.13
**SS3**	0.89 ± 0.02	8.25 ± 0.41	2.72 ± 0.03	8.26 ± 0.02	2.30 ± 0.03	1.91 ± 0.13	2.36 ± 0.05
**PS1**	0.68 ± 0.02	20.3 ± 0.7	< 0.02	<0.02	1.77 ± 0.02	0.45 ± 0.08	3.44 ± 0.10
**PS2**	0.84 ± 0.02	11.7 ± 0.6	2.45 ± 0.02	5.17 ± 0.09	2.15 ± 0.05	1.38 ± 0.24	2.13 ± 0.04
**A**	1.09 ± 0.02	37.0 ± 1.0	<0.02	0.49 ± 0.06	2.67 ± 0.05	1.23 ± 0.02	0.83 ± 0.10
**SeS**	1.54 ± 0.02	29.0 ± 0.6	2.67 ± 0.00	8.00 ± 0.30	9.32 ± 0.20	4.67 ± 0.05	1.07 ± 0.02
**MIN**	0.55	6.73	<0.02	< 0.02	1.15	<0.02	0.83
**MAX**	1.54	37.0	2.72	8.26	9.32	4.67	3.44
**MEAN**	0.81	16.12	0.79	2.12	2.61	1.25	1.75

## Data Availability

Data, associated metadata, and calculation tools are available from the corresponding author (paolo.inaudi@unito.it).

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
