# Peer review of "Analytical Methods for the Characterization of Vegetable Oils"

_molecules, 2022, doi:10.3390/molecules28010153_

Round 1

Reviewer 1 Report

The main objectives of this work, titled “Analytical methods for the characterization of vegetable oils” are to investigate the relationship between the inorganic components of oils, determined by ICP-OES, and their geographical origin, and to investigate the oxidation and reduction profile of different oils related to the presence of antioxidant compounds. The introduction is concise but complete. Material and methods are well explained, the analyses are accurate and validated from the use of a certified reference material. The work is interesting and deserve publication in a prestigious journal such as Molecules, and only minor revisions are required in accordance with the following recommendations.

Line 69-77. These sentences need references.

Table 1. Please insert the Note below the table to describe the classification reported.

Table 2. Please insert the Note to describe the reported codes.

Line 188, Replace “tip.” With “tip,”

Line 219-220. I suggest replacing “It is possible to see that percentage recoveries of more than 75% were obtained” with “Percentage recoveries from 75% to 101.5% were obtained”.

Line 223. Replace “the trueness” with a more scientific term “the accuracy”.

Line 252 (and 389). Replace “a few kilometers far from Italian coast” with “a few dozen kilometers far from the Italian coast”

Line 297-298. In figures S05-S08 authors reported only the cathodic scan of EVOOs, but in line 297 they wrote “and from one to two peaks in those associated with other oils”. This is a bit confusing: authors should also cite Figures related to SWV-C of VOs.

Line 296-306. I suggest to better describe the results obtained, citing the correct figures when appropriate.

Line 336. Replace “rise to changes in in chemical composition” with “rise to changes in chemical composition”.

Fig. 6. Describe in the caption the meaning of the acronyms given to oils. This would help the reader to visualize the results faster, without having to go back to Tables 1 and 2.

In Figures 7 and 8 the title of x axis overwrites the x axis scale. Moreover, the font size of the axis labels is smaller than in Figure 9. In Fig 7 the names of the region of origin (Spain, Umbria and Liguria) are lacking, as well as the circle that highlights the samples from Liguria region. In Figures 8 and 9 the name “Liguria” is not on the same line. Please check formatting.

Line 425. Author underlay for the first time the necessity of a stirring step. I suggest mentioning this fact before, moving the sentence in material and method. Moreover, in line 456-458 there is a repetition of the same concept.

The conclusion section is too long. Many sentences could be moved in the “Results and discussion” section. For example, the sentences from line 457 to line 454 should be put in the section of PCA. In general, I suggest shortening the conclusions, limiting to report the conclusions that can be drawn from the results obtained, while the evidences from the literature should be moved to the section “Results and discussion”.

Author Response

The main objectives of this work, titled “Analytical methods for the characterization of vegetable oils” are to investigate the relationship between the inorganic components of oils, determined by ICP-OES, and their geographical origin, and to investigate the oxidation and reduction profile of different oils related to the presence of antioxidant compounds. The introduction is concise but complete. Material and methods are well explained, the analyses are accurate and validated from the use of a certified reference material. The work is interesting and deserve publication in a prestigious journal such as Molecules, and only minor revisions are required in accordance with the following recommendations.

We thank the reviewer for the comment.

Special comments:

Line 69-77. These sentences need references.

Response: Thank you for your suggestions. We add two references (Morelló, J.-R.; Romero, M.-P.; Motilva, M.-J. Effect of the Maturation Process of the Olive Fruit on the Phenolic Fraction of Drupes and Oils from Arbequina, Farga, and Morrut Cultivars. J. Agric. Food Chem. 2004, 52, 6002–6009, doi:10.1021/jf035300p and Morelló, J.-R.; Motilva, M.-J.; Tovar, M.-J.; Romero, M.-P. Changes in Commercial Virgin Olive Oil (Cv Arbequina) during Storage, with Special Emphasis on the Phenolic Fraction. Food Chem. 2004, 85, 357–364, doi:10.1016/j.foodchem.2003.07.012) and we correct all the references.

Table 1. Please insert the Note below the table to describe the classification reported.

Response: Thank you for your suggestions, we now insert the Note below the table 1.

Table 2. Please insert the Note to describe the reported codes.

Response: Thank you for your suggestions, table 2 describes the reported codes.

Line 188, Replace “tip.” With “tip,”

Response: Thank you for your correction. We replace “tip.” with “tip,”.

Line 219-220. I suggest replacing “It is possible to see that percentage recoveries of more than 75% were obtained” with “Percentage recoveries from 75% to 101.5% were obtained”.

Response: Thank you for your suggestions, we now replace the sentence.

Line 223. Replace “the trueness” with a more scientific term “the accuracy”.

Response: Thank you for your suggestions, we now replace the sentence.

Line 252 (and 389). Replace “a few kilometers far from Italian coast” with “a few dozen kilometers far from the Italian coast”

Response: Thank you for your suggestions, we now replace the two sentences.

Line 297-298. In figures S05-S08 authors reported only the cathodic scan of EVOOs, but in line 297 they wrote “and from one to two peaks in those associated with other oils”. This is a bit confusing: authors should also cite Figures related to SWV-C of Vos.

Response: Thank you for your suggestions, we now replace the sentence with “Cathodic scans (Figures S05-S08) display from two to three peaks in the voltammograms relating to the EVOOs and from one to two peaks in those associated with other oils (Figures S17-S20).”

Line 296-306. I suggest to better describe the results obtained, citing the correct figures when appropriate.

Response: Thank you for your suggestions, we correct the sentence with “In VOs (Figures S13-S24) the peaks related to SWV-A are less numerous (up to three in SS2 and SS3, figures S13) and have lower height than in EVOOs. In all cases there is a low peak at 0.4 V (polyphenols), in sunflower and corn there are peaks around 0.6 V (polyphenols, figures S13-S14) and in SS2 and SS3 there is a third peak around 1.1 V (tocopherols, figure S17). In VOs, instead, there is always a peak at 0.4 V and in some samples a second peak appears: for CS1, CS2, PS2 (figures S14-S15) and all corn seed oils it is found at about 1.1 V, while for sesame oil it is observed near 0.2 V (figure S16 and S20).”

Line 336. Replace “rise to changes in in chemical composition” with “rise to changes in chemical composition”.

Response: Thank you for your suggestions, we delete “in”.

Fig. 6. Describe in the caption the meaning of the acronyms given to oils. This would help the reader to visualize the results faster, without having to go back to Tables 1 and 2.

Response: Thank you for your suggestions, we add all the acronyms in the caption.

In Figures 7 and 8 the title of x axis overwrites the x axis scale. Moreover, the font size of the axis labels is smaller than in Figure 9. In Fig 7 the names of the region of origin (Spain, Umbria and Liguria) are lacking, as well as the circle that highlights the samples from Liguria region. In Figures 8 and 9 the name “Liguria” is not on the same line. Please check formatting.

Response: Thank you for your correction, we check and correct all the figures.

Line 425. Author underlay for the first time the necessity of a stirring step. I suggest mentioning this fact before, moving the sentence in material and method. Moreover, in line 456-458 there is a repetition of the same concept.

Response: Thank you for your suggestions, we move “The stirring step is indispensable to homogenize each sample.” in material and method and we delete the repetition in the conclusions.

The conclusion section is too long. Many sentences could be moved in the “Results and discussion” section. For example, the sentences from line 457 to line 454 should be put in the section of PCA. In general, I suggest shortening the conclusions, limiting to report the conclusions that can be drawn from the results obtained, while the evidences from the literature should be moved to the section “Results and discussion”.

Response: Thanks for the suggestion, we deleted some sentences. We modify the conclusions:

“The first part of this work was aimed at characterizing EVOOs samples from the point of view of the inorganic profile. The content of twelve fourteen elements, namely Al, Ba, Ca, Cu, Fe, K, Li, Mn, Mg, Na, P, Sb, Se and Zn was quantified by ICP-OES, after pre-treatment of the samples by stirring and acid digestion. The stirring step is indis-pensable to homogenize each sample. The inorganic component was investigated in order to find out if it was possible to discriminate the different geographical origin of the samples from it.

The element concentrations present in the oils, in fact, depend directly on the ones that the plant absorbs and accumulates from the growing medium, i.e. on the composi-tion of the soil of cultivation, on the metabolic and therefore genetic, profile of the plant of origin as well as on the applied agronomic, extractive and storage techniques. Ca, Mg, Na and Se are the major elements (>1 mg kg-1). K, on the other hand, had a very variable trend. The concentrations of Li, P, Ba, Sb and Zn were always below the detection limit of the instrument.]  The data show that Italian EVOOs have greater Fe content than non-Italian ones, and Umbrian EVOOs show higher K but lower Na Al concentrations than Ligurian ones; the composition of the oils obtained from other plants is rather heterogeneous and no common trend can be identified. The chemometric treatment of the experimental results through a PCA confirmed the possibilities offered by the method with regard to discrimination of geographical origin on the basis of the proximity of the different scores, although some samples failed to meet expectations because of a higher concentration of Se. Unfortunately, the compositions of the soils in which plants were grown is not available, so it is not possi-ble to identify the source of this element. In particular, the behaviour of the two samples from Portugal shows the possibility to distinguish oils provided by different pro-ducers, if the soil and the other characteristics of the area cultivated by each producer differ from each other: in synthesis, a distinction at local scale is feasible. The joint examination of scores and loadings shows that the Italian EVOOs have greater amounts of K, Fe and Na than the other EVOOs and, in general, the EVOOs are characterized by a constant presence of K and Na compared to other oils. Among the oils from other plants, it was possible to highlight the peculiar behavior of almond and sesame, due to their particular concentrations of metals (for almond Ca is almost 37 mg kg-1, while for sesame all elements have high concentrations). This behavior could be due both to the different pedoclimatic conditions of growth of the species of origin and to their botanical characteristics. A geographic and varietal characterization is therefore possible even if the analysis is critical because of the complex organic matrix. In particular, we observed that the homogenization of the sample by stirring before mineralization is very important to obtain repeatable results. Thus, the elementary analysis of oils can be a means to eval-uate product authenticity, for example by detecting cuts with oils of different botanical species, or to determine the pollution from heavy metals, deriving from the soil in which olive trees were grown or from accidental contamination during the extraction or distribution phases. Hence, it can be a useful tool for monitoring the quality (from the point of view of contamination by metals) and the genuineness (understood as ab-sence of sophistication) of the oil itself. The purpose of the second part of this work was to determine the redox profile of chemical species, naturally present in the composition of vegetable oils by voltamme-try, through the use of CPE prepared by introducing the same oil samples in the paste as binders. The presence of species with antioxidant properties, such as polyphenols and tocopherols, was detected from the peaks displayed in the voltammograms. SW provides more intense peaks than CV. The peaks in the profiles of different oils are dif-ferent in shape, position or intensity, because they are associated with antioxidant molecules of different nature (even if they share common features, i.e. the range of po-tential in which they appear) and present in different concentrations, which in turn are associated to different composition of the species from which the oil derives. Therefore voltammetry allowed us to discriminate different types of oils according to their botanical origin, both through the simple observation of the voltammetric pro-files, and by chemometric treatment of the results by PCA. In particular, all samples have at least one peak for polyphenols, but EVOOs show higher and more defined peaks; similarly, tocopherols are present in higher quantities in EVOOs. Chemometric treatment with PCA confirmed that the different oils are separated quite well in the score plot according to their botanical origin and in some cases also for geographic origin. In fact, not only the Spanish EVOOs were distinguished from the Italian ones, but also the Umbrian samples were differentiated from Ligurian ones, while the separation between Spanish and Portuguese EVOO was not so efficient. As expected, non-EVOO samples were differentiated according to the plant spe-cies of origin. In particular, sesame seed oil is completely isolated from the other test samples: this finding suggests that this oil can be easily distinguished from the other ones by comparing their redox profile.  In turn, these differences in chemical composition affect both the organoleptic characteristics and the stability of the oils. The antioxidant molecules, in fact, can take part in oxidation reactions to which the vegetable oils are naturally subject, protecting them from lipoperoxidation but attributing also a bitter and pungent taste to the oil. A high content in metals involves a catalysis of the lipoperoxidation reactions that cause early rancidity of the oil.

In this sense, voltammetry can be used as a monitoring tool for the production and distribution chain of oil and derivatives, assessing their conservation status, as well as for the identification of possible sophistication with oils of different botanical origin that would vary the intensity and the position of the peaks proper to the EVOO.

In conclusion, voltammetry presents good potential from the standpoint of moni-toring the quality, state of conservation and the organoleptic properties of oils and their by-product. Possible future development of the study includes: for the inorganic profile, the analysis of both oils and the soil in which olives were grown; for the redox profile, the analysis of a higher number of samples, to better investigate the potential of the technique.

Reviewer 2 Report

The manuscript of Giacomino et al. focused on identification of markers in several extra virgin olive oil samples. Chemical elements were determined by ICP-OES after pretreatment by acid mineralization. Electrochemical properties of the samples have been investigated by voltammetry. A chemometric evaluation of the results was performed to possibly distinguish i) the region of provenience considering the inorganic profile, ii) the plant species from which each oil was obtained on the basis of the electric current profile registered during the voltammetric analysis.

Authors planned the sample preparation and analysis carefully, the work presented seems to be reproducible in any other similar laboratory.

The English of the manuscript is good. However, minor refinements should be done. For example, at line 104 it is better to write that the objective of the present manuscript was twofold. Some sentences are too long.

Introduction should be amended with adulteration techniques for olive oils.

It is strange to give additional information (lines 117-122) after presenting the objectives of the study

line 493: Authors claim that: A high content in metals involves a catalysis of the lipoperoxidation reactions that cause early rancidity of the oil.

Although I agree with this statement, the main metal ions involved are redox cycling ones found in living organisms such as Fe and Cu. Why determination of Cu was not intended by ICP-OES

After reading the manuscript, I was expecting from the Authors to give a recommendation about the best approach for discriminating between different olive oils and detecting adulteration. Would it be enough to perform only the voltametric analyses? Or are the two techniques complementary as suggested because of the link between lipid peroxidation and catalytic activity of redox cycling metal ions?

Minor comments:

Line 15: Do not start the sentence with an Arabic number. Write ten instead.

Line 16: Do not start a sentence with an element symbol. Write either Aluminium ((GB) or Aluminum (US) instead.

Line 24, line 37, : Do not start a sentence with an acronym. Alternatively, “The EVOO…

line 59 and throughout the manuscript:  Oleaceae and all other Latin names should  be typeset in Italics.

Author Response

The manuscript of Giacomino et al. focused on identification of markers in several extra virgin olive oil samples. Chemical elements were determined by ICP-OES after pretreatment by acid mineralization. Electrochemical properties of the samples have been investigated by voltammetry. A chemometric evaluation of the results was performed to possibly distinguish i) the region of provenience considering the inorganic profile, ii) the plant species from which each oil was obtained on the basis of the electric current profile registered during the voltammetric analysis.

Authors planned the sample preparation and analysis carefully, the work presented seems to be reproducible in any other similar laboratory.

Response: We thank the reviewer for the comment.

The English of the manuscript is good. However, minor refinements should be done. For example, at line 104 it is better to write that the objective of the present manuscript was twofold. Some sentences are too long.

Response: We thank the reviewer, we modify the sentence and we deleted some sentences.

Introduction should be amended with adulteration techniques for olive oils.

Response: We thank the reviewer, we modify the sentence “Adulteration is a serious issue for EVOOs and it may take place by mixing EVOO with oils of lower quality, for examples by soybean oil [13]”, we also add a new reference.

Fasciotti, M.; Pereira Netto, A.D. Optimization and Application of Methods of Triacylglycerol Evaluation for Characterization of Olive Oil Adulteration by Soybean Oil with HPLC–APCI-MS–MS. Talanta 2010, 81, 1116–1125, doi:10.1016/j.talanta.2010.02.006

It is strange to give additional information (lines 117-122) after presenting the objectives of the study

Response: Thanks for the suggestion, we modify the sentence “The results of chemical analysis were processed with principal components analysis (PCA) [44] to search for a fingerprint of origin of oils, resulting from the trace element determination and voltammetric analysis.”

line 493: Authors claim that: A high content in metals involves a catalysis of the lipoperoxidation reactions that cause early rancidity of the oil. Although I agree with this statement, the main metal ions involved are redox cycling ones found in living organisms such as Fe and Cu. Why determination of Cu was not intended by ICP-OES?

Response: Thanks very much for your very much for this comment; it was our forgetfulness. Also the concentrations of Mn and Cu were below the LOD of the ICP-OES, we mistakenly forgot to add them. The importance of Mn, Cu and Zn for reductive oxide phenomena are very important, generally these elements in olive oils are determined with ICP-MS because they are present in concentrations of a few µg kg-1. The purpose of this manuscript was to understand if it was possible to identify the origin of the EVOOs based on the content of macro-elements

After reading the manuscript, I was expecting from the Authors to give a recommendation about the best approach for discriminating between different olive oils and detecting adulteration. Would it be enough to perform only the voltametric analyses? Or are the two techniques complementary as suggested because of the link between lipid peroxidation and catalytic activity of redox cycling metal ions?

Response: Thanks for the question. The voltammetric technique can certainly be considered an excellent technique for screening of oils in order to assess the presence or absence of antioxidants in EVOO. Surely to have a more complete view of the EVOOs quality, the voltammetric screening has to be coupled with other analytical techniques that allow the quantification of the antioxidants.

Minor comments:

Line 15: Do not start the sentence with an Arabic number. Write ten instead.

Response: Thank you for your suggestions, we correct the sentence “Ten10 vegetable oils (VOs), eight8 Italian EVOOs and seven7 not Italian EVOOs were analyzed.”

Line 16: Do not start a sentence with an element symbol. Write either Aluminium ((GB) or Aluminum (US) instead.

Response: Thank you for your suggestions, we modify the sentence “After pretreatment by acid mineralization, Al, Ba, Ca, Cu, Fe, K, Li, Mg, Mn, Na, P, Sb, Se and Zn were determined by ICP-OES”.

Line 24, line 37: Do not start a sentence with an acronym. Alternatively, “The EVOO…

Response: Thank you for your suggestions, we modify the sentence “The EVOO-CPEs..” and “The EVOO is considered..”

line 59 and throughout the manuscript:  Oleaceae and all other Latin names should  be typeset in Italics.

Response: Thank you for your suggestion, we write in Italics all the Oleaceae and all other Latin names, line 59 “Oleaceae” , line 61 “Olea Europaea L.”, line 62 “Olea Europaea Var. Sylvestris (Mill.) lehr
